# Cryo-EM structure of the yeast TREX complex and coordination with the SR-like protein Gbp2

Yihu Xie[1]*, Bradley P Clarke[1], Yong Joon Kim[2,3], Austin L Ivey[1], Pate S Hill[1], Yi Shi[2,3], Yi Ren[1]*

[1]Department of Biochemistry, Vanderbilt University School of Medicine, Nashville, United States; [2]Department of Cell Biology, University of Pittsburgh, Pittsburgh, United States; [3]Medical Scientist Training Program, University of Pittsburgh and Carnegie Mellon University, Pittsburgh, United States

**Abstract** The evolutionarily conserved TRanscript-EXport (TREX) complex plays central roles during mRNP (messenger ribonucleoprotein) maturation and export from the nucleus to the cytoplasm. In yeast, TREX is composed of the THO sub-complex (Tho2, Hpr1, Tex1, Mft1, and Thp2), the DEAD box ATPase Sub2, and Yra1. Here we present a 3.7 Å cryo-EM structure of the yeast THO•Sub2 complex. The structure reveals the intimate assembly of THO revolving around its largest subunit Tho2. THO stabilizes a semi-open conformation of the Sub2 ATPase via interactions with Tho2. We show that THO interacts with the serine–arginine (SR)-like protein Gbp2 through both the RS domain and RRM domains of Gbp2. Cross-linking mass spectrometry analysis supports the extensive interactions between THO and Gbp2, further revealing that RRM domains of Gbp2 are in close proximity to the C-terminal domain of Tho2. We propose that THO serves as a landing pad to configure Gbp2 to facilitate its loading onto mRNP.

## Introduction

Eukaryotic RNA transcription is carried out in the nucleus by the RNA polymerases. During an early stage of mRNA transcription, a 5' cap is added to the newly synthesized mRNA, which is followed by splicing, 3'-end processing, and polyadenylation. Nuclear mRNA biogenesis culminates in their export through the nuclear pore complex to the cytoplasm. Many protein factors including serine-arginine (SR) proteins associate with mRNAs to form mature messenger ribonucleoproteins (mRNPs) for export (*Metkar et al., 2018*; *Singh et al., 2012*). The evolutionarily conserved TRanscript-EXport (TREX) complex plays key roles in the highly coordinated mRNP assembly and export (*Carmody and Wente, 2009*; *Chávez et al., 2000*; *Luo et al., 2001*; *Strässer and Hurt, 2001*; *Strässer et al., 2002*; *Viphakone et al., 2019*; *Xie and Ren, 2019*; *Zhou et al., 2000*). TREX is recruited to actively transcribed genes (*Cheng et al., 2006*; *Masuda et al., 2005*; *Strässer et al., 2002*) and impacts transcription especially during elongation (*Domínguez-Sánchez et al., 2011*; *Zhang et al., 2016*).

The C-terminal domain of the largest subunit of RNA Pol II is highly phosphorylated on the hepta-peptide repeats (YSPTSPS) at the Serine 2 position during the elongation phase of the transcription cycle (*Hsin and Manley, 2012*). Serine 2 phosphorylation coordinates loading of co-transcriptional 3'-end processing factors to the transcription machinery (*Ahn et al., 2004*). In yeast, the primary RNA Pol II CTD Ser2 kinase is the CTDK-1 complex (*Cho et al., 2001*; *Sterner et al., 1995*; *Wood and Shilatifard, 2006*). Growing evidence links the function of TREX and transcriptional CDKs. The yeast TREX component Mft1 interacts genetically with CTDK-1 (*Hurt et al., 2004*). In addition to their roles during transcription elongation, TREX and CTDK-1 both influence mRNA 3'-end processing and polyadenylation (*Ahn et al., 2004*; *Rougemaille et al., 2008*; *Saguez et al.,*

*For correspondence:
yihu.xie@vanderbilt.edu (YX);
yi.ren@vanderbilt.edu (YR)

Competing interests: The authors declare that no competing interests exist.

*2008*). In humans, the transcriptional kinases are more divergent, at least CDK9, CDK11, CDK12, and CDK13 are shown to phosphorylate Ser2 on Pol II CTD, among which CDK12 and CDK13 are orthologs of yeast CTDK-1. These human CDKs are recognized as potential targets for cancer therapy (*Cao et al., 2014*; *Parua and Fisher, 2020*). TREX and CDK11 have been shown to interact in human cells and play roles in regulating HIV mRNA 3'-end processing (*Pak et al., 2015*).

The coordination of TREX and CTDK-1 is largely unknown. Several lines of evidence suggest that a group of shuttling SR proteins could serve as the link for THO and CTDK-1. SR proteins are well recognized as splicing factors, but they also play important roles in coordinating transcription and mRNA export (*Reed and Cheng, 2005*). In yeast, there are three shuttling SR-like proteins, Gbp2, Hrb1, and Npl3, which play roles in mRNA export by interacting with the mRNA export receptor Mex67•Mtr2 (*Hackmann et al., 2014*). In humans, three SR proteins, SRSF1, SRSF3, and SRSF7, also shuttle between the nucleus and the cytoplasm to facilitate mRNA export by serving as adaptors for the human ortholog of Mex67•Mtr2, the NXF1•NXT1 complex (*Huang et al., 2003*; *Huang and Steitz, 2005*; *Müller-McNicoll et al., 2016*).

In yeast cells, TAP-tagged Gbp2 and Hrb1 were shown to associate with the CTDK-1 complex (*Hurt et al., 2004*). Consistent with this observation, using purified recombinant proteins, we recently showed that Gbp2 RRM domains are sufficient to interact with CTDK-1, involving the N-terminal RS domain in its Ctk1 kinase subunit (*Xie et al., 2021*). We also found that there is a synthetic growth defect when both CTK1 and GBP2 are knocked out in yeast. The physical and functional interactions between Gbp2 and CTDK-1 provide a link between Gbp2 function and the transcription machinery. Interestingly, in humans, CDK11 interacts with SRSF7 (*Hu et al., 2003*), and together with TREX, all are implicated in HIV-1 mRNA 3'-end processing (*Pak et al., 2015*; *Valente et al., 2009*). Among the three yeast shuttling SR-like proteins, Gbp2 and Hrb1, but not Npl3 have been shown to rely on the THO components Hpr1 and Mft1 to load onto mRNPs (*Häcker and Krebber, 2004*). The different requirements could stem from an interaction between THO and Gbp2 and Hrb1, but not Npl3 (*Hurt et al., 2004*; *Martínez-Lumbreras et al., 2016*).

Despite extensive studies, how TREX, SR proteins, and CTDK-1 coordinately function during mRNA biogenesis is still not clear. To elucidate the molecular mechanisms, we conducted biochemical and structural studies on the yeast TREX complex and Gbp2. Yeast TREX is a ~470 kDa protein complex comprised of the pentameric THO sub-complex (Tho2, Hpr1, Tex1, Mft1, and Thp2), the DEAD box ATPase Sub2, and Yra1. Thus far structural understanding of the TREX complex has been limited to low resolution structures (*Peña et al., 2012*; *Ren et al., 2017*). Here we present a 3.7 Å cryo-EM structure of the yeast THO•Sub2 complex to reveal the molecular details of the THO complex assembly and the THO–Sub2 interactions. We demonstrate direct binding between THO and Gbp2 using recombinant proteins and dissect their mode of interaction using in vitro binding studies and cross-linking mass spectrometry (XL-MS) analysis of the THO–Gbp2 complex. Together, we propose that TREX serves as a landing pad to configure the multi-domain Gbp2 and facilitate its loading onto the mRNP.

## Results and discussion

### THO directly interacts with the SR-like protein Gbp2

We began by testing the interaction between the THO complex and Gbp2 using purified recombinant proteins. The ~400 kDa THO complex consisting of full-length Tho2, Hpr1, Tex1, Mft1, and Thp2 subunits (denoted by THO–FL, *Figure 1A*) was expressed in insect cells. Full-length Gbp2 was expressed in insect cells with an N-terminal GST-tag. Using GST pull down assays, we show that Gbp2 directly interacts with THO–FL (*Figure 1B*). We next tested the binding of Gbp2 to a THO core complex (denoted by THO*, *Figure 1A*) that contains the ordered regions of all THO's five subunits. We found that THO* is capable of binding to Gbp2, but with reduced interaction compared to THO–FL (*Figure 1B*). These results suggest that multiple regions in THO are involved in Gbp2 recognition, including both the THO core and the potentially flexible regions that are truncated in THO*.

We next attempted to dissect the domains in Gbp2 that are involved in THO interaction. Gbp2 contains an N-terminal RS domain followed by three tandem RRM domains, RRM1, RRM2, and RRM3 (*Figure 1A*). RRM1 and RRM2 domains are capable of binding to RNA. RRM3 was shown to recognize THO (*Martínez-Lumbreras et al., 2016*). Interestingly, we found that Gbp2 without RRM3

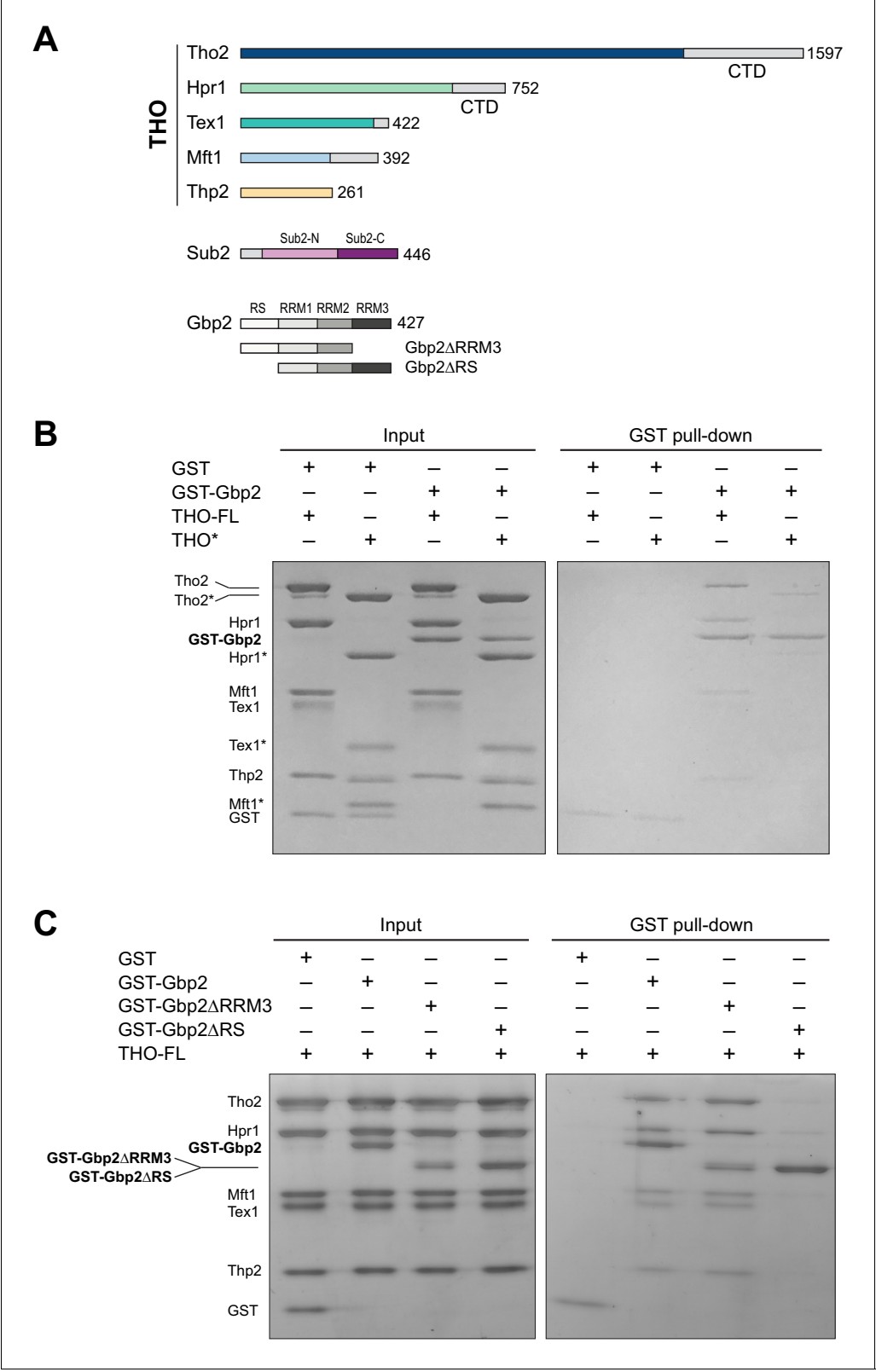

**Figure 1.** The THO complex directly interacts with the SR-like protein Gbp2. (**A**) Domain organization of the THO complex, Sub2, and Gbp2. Within THO, the protein regions that are included in the core THO* complex are colored (Tho2 in blue, Hpr1 in green, Tex1 in cyan, Mft1 in light blue, and Thp2 in yellow). Sub2 is colored in pink (Sub2-N) and purple (Sub2-C). Gbp2 contains an N-terminal RS domain followed by three RRM domains. (**B**)

*Figure 1 continued on next page*

*Figure 1 continued*

THO directly interacts with Gbp2. In vitro GST-pull down assays with purified recombinant proteins show that both THO–FL and THO* bind to Gbp2 with the former exhibiting stronger interaction. (C) THO binding to Gpb2 requires the N-terminal RS domain of Gbp2.

still binds to THO (*Figure 1C*). On the other hand, deletion of the N-terminal RS domain of Gbp2 substantially reduced THO interaction, suggesting that the Gbp2 RS domain is required for stable binding to THO.

Together, our binding studies indicate that THO–Gbp2 interaction involves multiple domains from both THO and Gbp2. To provide insights into the underlying molecular mechanisms of the THO–Gbp2 recognition, we take an integrative approach combining cryo-EM structure determination of the THO* core complex and XL-MS analysis of the THO–FL interaction with Gbp2.

## Cryo-EM structure of the THO*•Sub2 complex at 3.7 Å resolution

The THO complex is an integrated structural and functional unit that regulates the activity of Sub2. We previously determined a THO•Sub2 crystal structure at 6.0 Å resolution (*Ren et al., 2017*). Here, we carried out single particle cryo-EM studies on THO*•Sub2. For cryo-EM sample preparation, the THO*•Sub2 complex was subjected to cross-linking with glutaraldehyde to obtain a more homogeneous sample. Full length THO complex isolated from yeast cells was shown to form a dimeric assembly by negative stain EM (*Peña et al., 2012*). We observed a higher ordered assembly composed of four THO*•Sub2 protomers (*Figure 2—figure supplement 1*, *Figure 2—figure supplement 2*, and *Supplementary file 1*). This THO*•Sub2 tetramer can be dissected as two dimers related by twofold symmetry, Dimer 1 (protomer 1A and 1B) and Dimer 2 (protomer 2A and 2B). Within each dimer, a coiled-coil region (corresponding to Mft1 and Thp2) from each protomer interact via a 'head on' mode. The interaction between the two dimers is mediated by the same coiled-coil region via a 'side to side' mode involving protomer 1A and 2A. It is interesting to note that the recent structure of human THO in association with UAP56, the human ortholog of Sub2, also exhibits a tetrameric assembly (*Pühringer et al., 2020*). The 'head on' mode interaction is conserved between yeast THO•Sub2 and human THO•UAP56 (*Figure 2—figure supplement 2A and B*). However, tetramerization of human THO•UAP56 is mediated by the Thoc6 subunit and a Thoc5 domain that do not exist in yeast. Our XL-MS analysis of the full length THO in association with Gbp2 (details in later section) identified multiple cross-links between a 'bulge' (Mft1 a.a. 142–196) and Tho2 C-terminal region (*Figure 2—figure supplement 2C* and *Supplementary file 2*) that can only be explained by the interface between two protomers within a dimer, whereas no cross-links support the 'side to side' interface of the tetramer. Of note, the THO* core contains ~600 residues less than the full length THO complex. Given that both full length THO isolated from yeast cells (*Peña et al., 2012*) and the full length THO in association with Sub2 purified from recombinant proteins (*Schuller et al., 2020*) exhibit a dimer, we speculate that the THO*•Sub2 tetramer was formed due to the truncations in THO*. Therefore, we focus on the THO*•Sub2 dimer in this study.

Each THO*•Sub2 dimer contains a rigid protomer (1A or 2A) and a mobile protomer (1B or 2B) (*Figure 2—figure supplement 2C*). This dimer observed by cryo-EM is consistent with our previously determined crystal structure which contains one THO*•Sub2 and a second THO*•Sub2 that is only partially resolved (*Figure 2—figure supplement 2D*; *Ren et al., 2017*). Comparison of the THO*•Sub2 dimer in our work and the recently reported structures of yeast THO•Sub2 and human THO•UAP56 reveals that the relative orientation between the protomers is flexible (*Figure 2—figure supplement 2E*; *Pühringer et al., 2020*; *Schuller et al., 2020*). For obtaining the best quality map for model building, the THO*•Sub2 protomer was extracted from the two rigid copies (1A and 2A) within the tetramer and refined to an overall resolution at 3.7 Å (*Figure 2—figure supplement 1* and *Supplementary file 1*). The electron density map allows us to build an atomic model of the THO complex de novo (*Figure 2A* and *Figure 2—figure supplement 3*). The THO model contains 2000 residues with 90% assigned residue register. Sub2 was modeled using our previously determined crystal structure (*Ren et al., 2017*). By having the resolution to build an atomic model, we now reveal the molecular details of the structural core of THO and its interaction with Sub2.

The structure of the THO complex reveals intimate folding of the five subunits (*Figure 2B* and *Figure 2—figure supplement 3*). Tho2, the largest subunit spanning the entire length of the

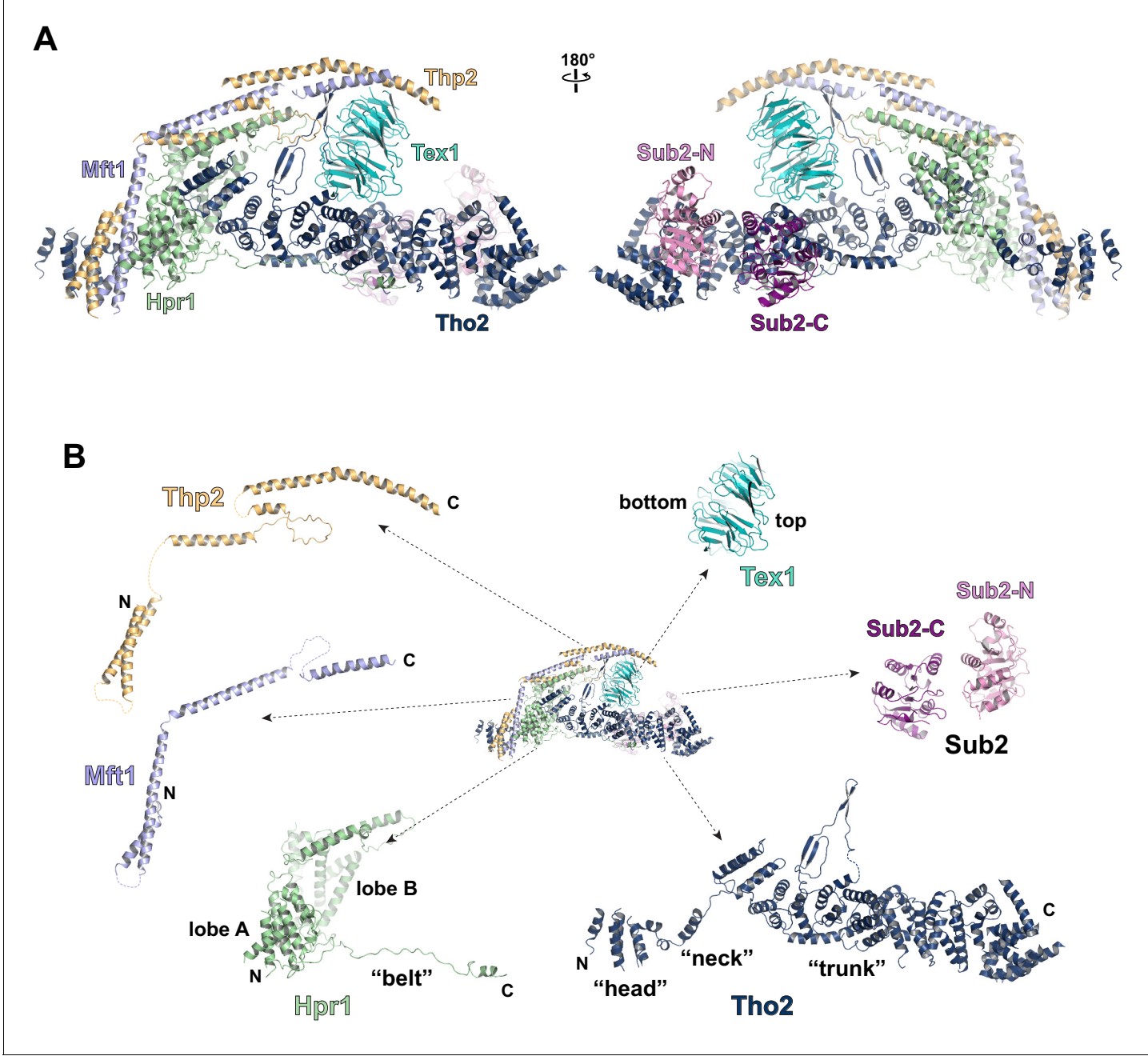

**Figure 2.** Cryo-EM structure of the THO*•Sub2 complex at 3.70 Å resolution. (**A**) Overall architecture of the THO*•Sub2 complex in front and back views. (**B**) Dissected view of the THO*•Sub2 complex subunits. The largest THO subunit, Tho2, contains a 'head', a 'neck', and an α-solenoid 'trunk'. Hpr1 contains lobe A, lobe B, followed by an extended 'belt'.

The online version of this article includes the following figure supplement(s) for figure 2:

**Figure supplement 1.** Cryo-EM data processing.

**Figure supplement 2.** Structural comparison of yeast and human TREX.

**Figure supplement 3.** Cryo-EM model building.

**Figure supplement 4.** Sequence alignment of Tho2 homologues.

elongated THO, plays a critical role in THO assembly. Tho2 can be dissected as 'head', 'neck', and 'trunk' sections. The Tho2 'head' contains an N-terminal helical bundle that clusters with the N-termini of Hpr1, Mft1, and Thp2. Tho2 'neck' is comprised of a helix followed by a loop. The 'neck' is embraced by a bi-lobed Hpr1 (lobe A and lobe B). Tho2 'trunk' folds into an alpha solenoid structure, which binds the Tex1 β-propeller at its center and stabilizes a semi-open Sub2 ATPase at its C-terminal end. An extended region at Hpr1 C-terminal region forms a 'belt' lining the Tho2 'trunk'.

## Assembly of the THO•Sub2 complex

Tho2 is the main scaffold upon which other THO constituents assemble (*Figure 2—figure supplement 4*). Tho2 features a total contact area of ~9000 Å$^2$ with the other four THO subunits. Tho2 'head' domain binds to a four helix bundle, formed by two pairs of anti-parallel helices contributed by Mft1 and Thp2, respectively (*Figure 3A*). Tho2 'head' and the helix in its 'neck' sandwich the very N-terminal helix of Mft1 (residues 6–17). The opposite side of the Mft1/Thp2 four helix bundle runs in parallel with Hpr1 lobe A (residues 1–230). The Tho2 'neck', particularly the loop (residues 167–179), is embraced by the Hpr1 lobe A and lobe B (residues 250–490) (*Figure 3B*). Although the 'neck' is largely buried, it contains multiple hydrophilic residues including K171, N173, and E177. Tho2 and Hpr1 residues at this interface are highly conserved from yeast to human (*Figure 2—figure supplement 4*).

The 'trunk' of Tho2 (residues 180–1200) forms an alpha-solenoid. Hpr1 'belt' contains residue assignment from residues 491 to 535 (*Figure 3C*). It starts from the beginning of the Tho2 'trunk', featuring aromatic residues at the interface including F511, F515, F518, and W532, and likely extends further to the C-terminus of Tho2 'trunk' as evidenced by our XL-MS studies discussed later. The seven-bladed Tex1 β-propeller sits at the center of the Tho2 'trunk' via blades 4 and 5 (*Figure 3D*). The loops connecting blades 4/5 (4D5A) and 5/6 (5D6A), as well as the 5BC loop within blade five contact a pair of Tho2 helices (residues 626–666), whose opposite side binds to the C-terminal RecA domain of Sub2 (Sub2-C). This Tho2–Tex1 interaction is conserved from yeast to human based on the sequence homology (*Figure 2—figure supplement 4*). In addition, a prominent extension from Tho2 is projected outward perpendicular to the Tho2 'trunk'. The C-terminal part of this extension (residues 464–485) forms a hairpin that winds through the bottom face of the Tex1 β-propeller. This additional Tho2–Tex1 binding mode is likely a yeast-specific mechanism as human and other metazoan THOs lack this extension (*Figure 2—figure supplement 4*).

Regulation of the enzymatic activity of the DEAD-box ATPase is vital to the stepwise remodeling reactions mediated by the TREX complex (*Xie and Ren, 2019*). We previously showed that THO stimulates the ATPase activity of Sub2 in vitro (*Ren et al., 2017*). The cryo-EM structure provides new insights into the molecular details of their interaction. Overall, THO stabilizes a semi-open conformation of Sub2 by interacting with both RecA domains (Sub2-N and Sub2-C). Comparison of the cryo-EM structure and our previous THO•Sub2 crystal structure shows that these two structures are in excellent agreement (*Figure 3—figure supplement 1*). The cryo-EM structure reveals the atomic details of the THO–Sub2 interactions at the Sub2-C interface (*Figure 3E*). Sub2-C makes contacts with two pairs of Tho2 helices (residues 625–695). The Sub2 loop consisting of residues 304–308 is situated at the center of the interface featuring electrostatic interactions via E305 and N307. In addition, another Sub2 loop consisting of residues 355–358 makes critical contacts via F355 and R358. The importance of this loop is evidenced by our previous mutagenesis studies that show the ATPase activity of Sub2 mutant E356A/K357A/R358A cannot be activated by THO (*Ren et al., 2017*). Sub2 assumes a closed conformation in its enzymatic active state, in which Sub2-N and Sub2-C clamp onto its RNA substrate. As we previously illustrated (*Ren et al., 2017*), the semi-open Sub2 is more similar to its active state than free Sub2 in which Sub2-N and Sub2-C are likely separated. In vitro, the observed stimulation of Sub2 activity may reflect change in dynamics of switching between different conformational states as a result of THO binding. In cells, Sub2 is recruited to the transcription machinery via THO and the semi-open Sub2 may represent a 'primed' state that enables efficient transition to the active state when a physiological RNA substrate is encountered. This Sub2 activation mechanism is a conserved mechanism shared by several other DEAD-box proteins including Dbp5 which functions at the terminal step of nuclear mRNA export at the cytoplasmic side of the nuclear pore complex (*Folkmann et al., 2011*; *Mathys et al., 2014*; *Montpetit et al., 2011*; *Schütz et al., 2008*).

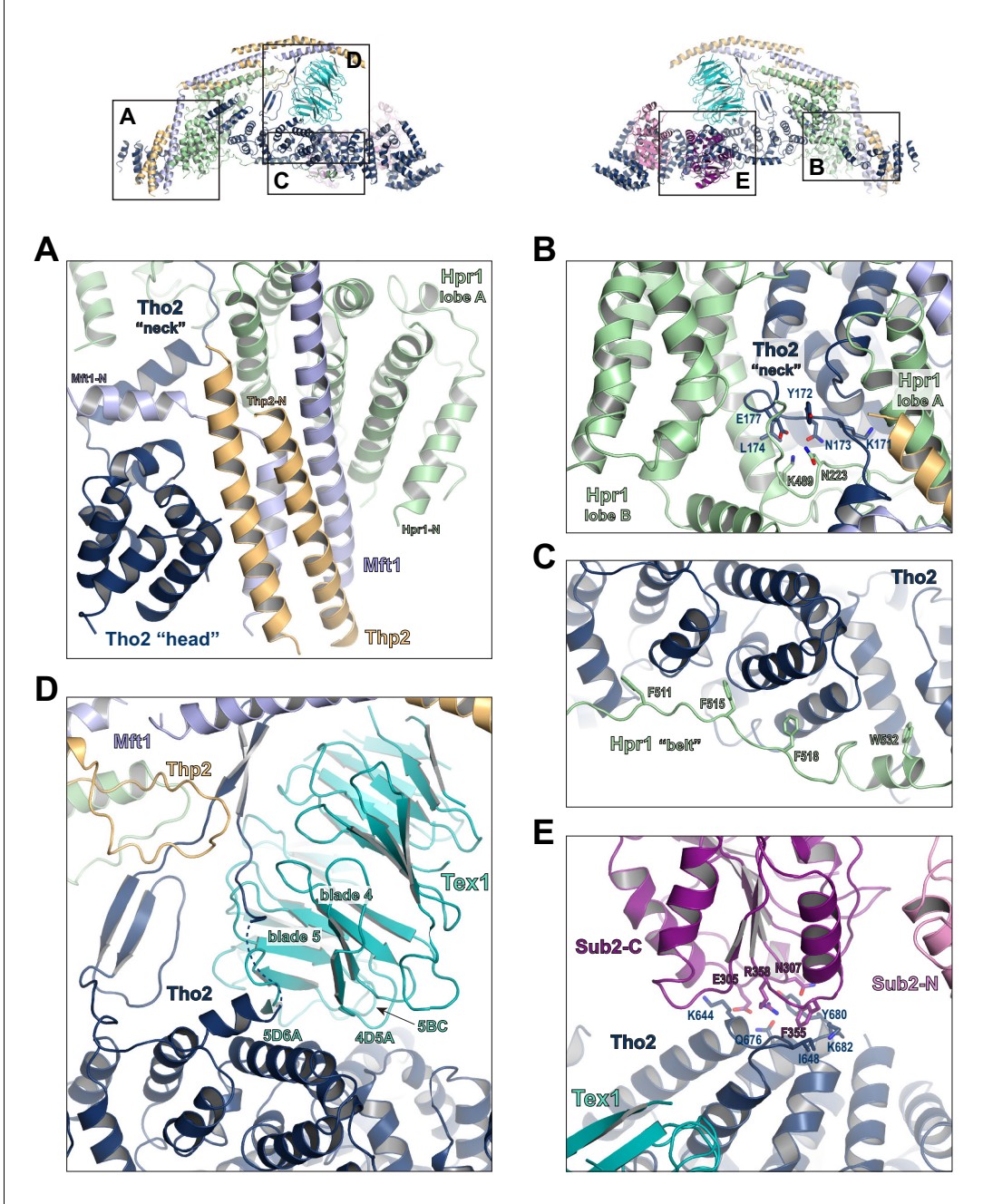

**Figure 3.** Key interactions in the THO*•Sub2 complex. (A) A highly intimate interface involving the Tho2 'head'. (B) The Tho2 'neck' is embraced by the two lobes of Hpr1. (C) The Hpr1 exhibits an extended 'belt' lining the Tho2 'trunk'. (D) The Tex1 beta propeller sits at the center of the Tho2 'trunk'. (E) The interface between Tho2 and Sub2-C.

The online version of this article includes the following figure supplement(s) for figure 3:

**Figure supplement 1.** Comparison of the cryo-EM structure and our previous crystal structure of THO•Sub2 at the THO–Sub2 interface.

## XL-MS analysis of the THO•Gbp2 complex

The THO complex contains a significant amount of potentially flexible regions including ~400 residues at the Tho2 C-terminal end and ~150 residues at the Hpr1 C-terminal end. These flexible regions are presumably not suitable for structural studies. Our binding studies show that these flexible regions are involved in Gbp2 recognition (*Figure 1B*). To gain further insights into the THO complex arrangement and the THO–Gbp2 interaction, we took a XL-MS approach (*Chait et al., 2016;*

*Leitner et al., 2016*; *Yu and Huang, 2018*) to analyze the complex between THO–FL and Gbp2. We used both EDC and DSS, a carboxyl and amine-reactive cross-linker and an amine-specific cross-linker that cross-link residues with Cα-Cα distance less than 17 Å and 30 Å, respectively (*Kim et al., 2018*; *Shi et al., 2014*). We obtained a total of 200 unique EDC cross-links, of which 69 were inter-protein cross-links including nine cross-links between Tho2 and Gbp2. We also obtained a total of 133 unique DSS cross-links to complement the EDC cross-link data, with 53 of these cross-links being interprotein cross-links (*Figure 4A*, *Figure 4—figure supplement 1A*, and *Supplementary file 2*). The cross-linking data is highly consistent with the THO structure (*Figure 4—figure supplement 1B and C*). 91% and 100% of the EDC and DSS cross-links that can be mapped to the structure fall within the expected distance restraint.

As previously mentioned, our XL-MS data provide experimental evidence to support the 'head on' THO–THO dimer (*Figure 2—figure supplement 2C*). In particular, we identified multiple cross-links involving the Mft1 'bulge' and the C-terminal region of Tho2 (Mft1-K182/Tho2-K1103, Mft1-K165/Tho2-K967, Mft1-K170/Tho2-K967, and Mft1-K174/Tho2-K967) (*Supplementary file 2*). These cross-links unambiguously point to the THO–THO interface within the 'head on' dimer. Furthermore, cross-links between THO subunits provide insights into the C-terminal domain of THO (Tho2–CTD, residues 1200–1597) downstream of the 'trunk' domain and the role it plays on the arrangement of the THO–THO dimer. The Tho2–CTD contains a 'bridge' that connects THO to the neighboring THO molecule as indicated by our cryo-EM density map (*Figure 4B*, *Figure 2—figure supplement 2C*). Comparison with the recently published THO–Sub2 structure reveals that the bridge starts at Tho2 residue 1200 (*Schuller et al., 2020*). The 'bridge' is followed by a structured segment, as suggested by the clustered cross-linking between Tho2 (residues 1260–1369) and the Hpr1 lobe B (E297, D434, K462, and K467) as well as Mft1 D129 (*Figure 4B*). In line with our observation, Tho2 (residues 1279–1405) was shown to form a rigid core through proteolysis and it folds into a helical structure as indicated by CD spectra (*Peña et al., 2012*). Importantly, cross-linking involving the structured segment indicates that the Tho2–CTD crosses over to the neighboring THO near its Hpr1 lobe B. The structured segment is followed by a highly flexible tail (residues ~1400–1597), as this region cross-links to spatially separated residues. For example, Tho2 K1576 cross-links to both Hpr1 lobe B (E297 and D434) and Tex1 (D341). In support of the flexibility of the Tho2 tail, a previous study showed that Tho2 (1411–1530) was highly sensitive to trypsin digestion (*Peña et al., 2012*). Both our THO•Sub2 structure and that recently published by others (*Schuller et al., 2020*) capture an asymmetric dimer exhibiting significant flexibility in the relative orientation of two THO•Sub2 molecules (*Figure 4—figure supplement 2*). The 'bridge' is only observed at the proximal side of the THO dimer. It is conceivable that the Tho2–CTD will exhibit more significant flexibility at the distal side of the THO dimer. Our data also provide insights into the arrangement of the Tex1 C-terminal tail (residues 367–422) and Hpr1–CTD (residues 600–752) (*Figure 4—figure supplement 2*). The extensive cross-links observed between Tho2–CTD and Hpr1–CTD suggests that they are spatially close to each other and are likely localized in between two THO molecules. Together, XL-MS results provide critical insights into the regions in THO that are not visible in the cryo-EM structures.

Cross-linking between Tho2 and Gbp2 indicates that Tho2–CTD is in close proximity to all three Gbp2 RRM domains (*Figure 4A and C*). Each of the three RRM domains cross-links to the structured segment in Tho2–CTD: RRM1-K190 to Tho2-K1349, RRM2-E241 to Tho2-K1250, and RRM3-D367 to Tho2-K1335. These results suggest that Gbp2 is localized in between two THO molecules near Hpr1 lobe B, as these involved Tho2 residues (K1250 and K1335) are cross-linked to Hpr1 lobe B (*Figure 4B*). Our data also show that each RRM domain cross-links to the highly flexible tail in Tho2–CTD. It is possible that, in the presence of Gbp2, the Tho2 tail may assume a more specific conformation.

Our XL-MS results (*Figure 4C*), together with the in vitro binding studies (*Figure 1B and C*), demonstrate that Tho2–CTD contributes to Gbp2 interaction. The C-terminal domain of Tho2 also binds to RNA/DNA (*Peña et al., 2012*). The function of Tho2–CTD in vivo was supported by the growth defect of *tho2-ΔCTD* yeast strains (*Peña et al., 2012*). Importantly, the synthetic growth defect of *tho2-ΔCTD* and *Δgbp2* strains highlights their functional links (*Martínez-Lumbreras et al., 2016*).

As both Gbp2 and Sub2 bind to the C-terminal region of Tho2, we next asked whether Gbp2 and Sub2 can associate with the THO complex together. GST–Gbp2 was used to pull down THO in the presence of Sub2. We found that GST–Gbp2 is able to pull down both THO and Sub2, and THO and Sub2 appear to be in a stoichiometric amount relative to each other (*Figure 4D*). In addition, GST–

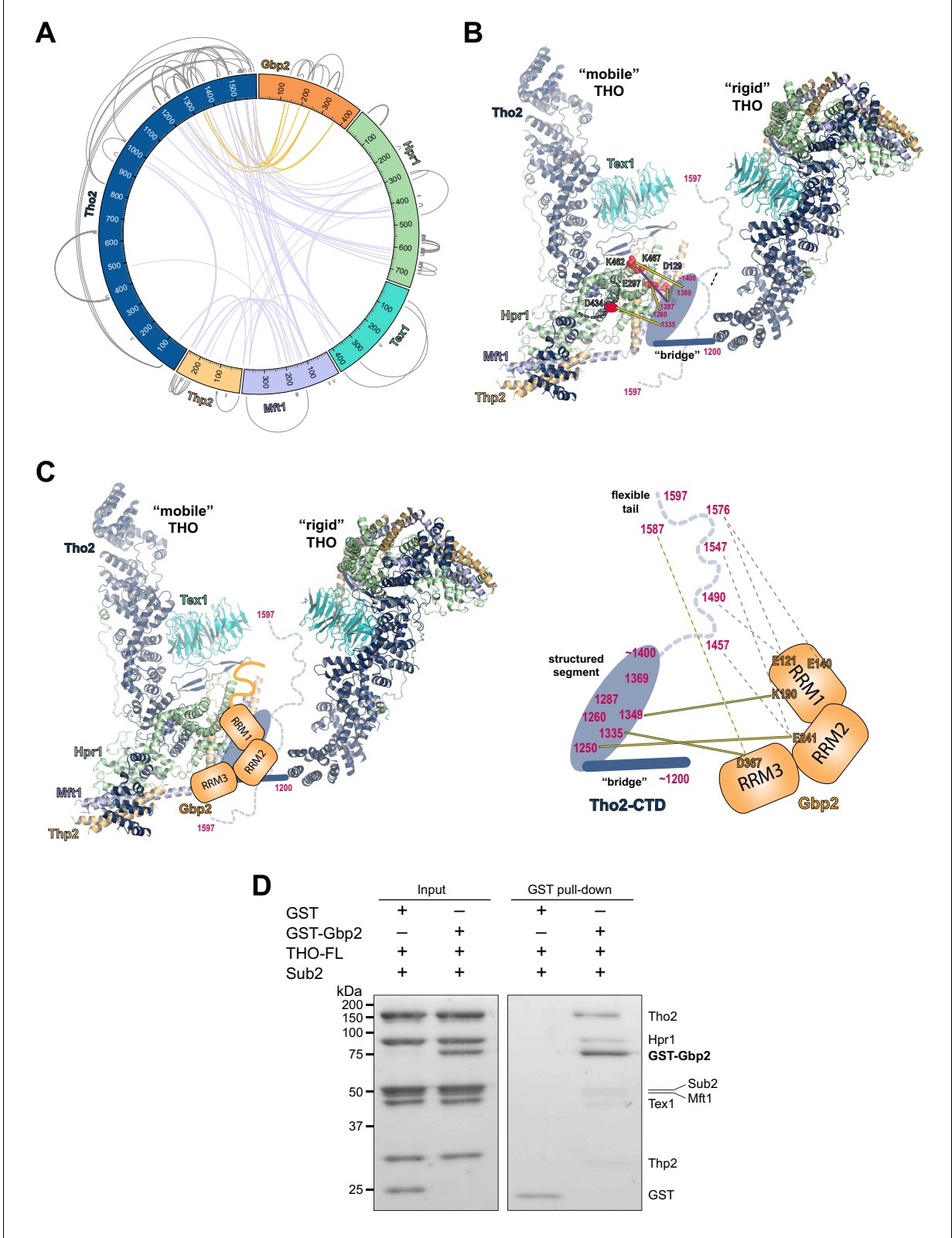

**Figure 4.** Chemical cross-linking and mass spectrometry reveals THO–Gbp2 interactions. (**A**) Circular plot showing the cross-linking sites with EDC cross-linker. Each THO•Gbp2 complex subunit is represented as a colored segment with the amino acid residues indicated. Intermolecular cross-links are mapped inside the circle and the intramolecular cross-links are mapped outside the circle. The cross-links between Tho2 and Gbp2 are colored in orange. (**B**) Schematics of the arrangement of the Tho2–CTD, which contains a 'bridge' connecting two THO molecules, followed by a structured

*Figure 4 continued on next page*

*Figure 4 continued*

segment and a flexible tail (residues ~1400–1597). The EDC cross-links between the structured Tho2–CTD fragment and Hpr1 (E297, D434, and K462) as well as Mft1 (D129) are indicated by yellow lines. The DSS cross-link between Tho2–CTD and Hpr1–K467 is indicated by a purple line. (**C**) Schematics of the THO–Gbp2 interactions (left) and the identified cross-linking sites between Tho2–CTD and Gbp2 RRM domains. (**D**) In vitro GST-pull downs show that Gbp2 binds to the THO•Sub2 complex.

The online version of this article includes the following figure supplement(s) for figure 4:

**Figure supplement 1.** Analyses of the XL-MS data.

**Figure supplement 2.** XL-MS data indicate the arrangement of the C-termini of Tex1 and Hpr1.

**Figure supplement 3.** Gbp2 does not interact with Sub2.

Gbp2 alone does not make direct contact with Sub2 (*Figure 4—figure supplement 3*). Our results suggest that THO, Sub2, and Gbp2 can form a THO•Sub2•Gbp2 complex, and therefore Gbp2 could function together with the TREX complex during nuclear mRNP maturation.

## Working model for coordinated function of TREX and Gbp2

Together with our recent characterization of Gbp2 interaction with the RNA Pol II Ser2 kinase CTDK-1 complex, we propose a working model for the coordinated function between TREX, Gbp2, and CTDK-1 (*Figure 5*). Gbp2 interaction with CTDK-1 provides a means to associate with the transcription machinery (*Hurt et al., 2004*; *Xie et al., 2021*). We envision that TREX and Gbp2 function coordinately during nuclear mRNP maturation and surveillance. During transcription, faulty assembly of mRNPs is a threat to genomic stability. If the defective mRNPs persist, they need to be sensed by a surveillance system and degraded. In yeast, Gbp2 and Hrb1 were shown to play key roles in mRNP surveillance (*Hackmann et al., 2014*). Interactions between Gbp2 and Mex67 for export and between Gbp2 and Mtr4 for degradation through the RNA exosome complex are mutually exclusive. TREX travels with the transcription machinery (*Meinel et al., 2013*) and its function in mRNP assembly is well documented. In THO/Sub2 mutant yeast cells, mRNP assembly is defective and faulty mRNPs cannot be degraded efficiently, which leads to the formation of heavy chromatin (*Rougemaille et al., 2008*; *Saguez et al., 2008*). In humans, depletion of TREX complex components leads to R-loop accumulation, transcriptional elongation defects, and trapped mRNP in nuclear

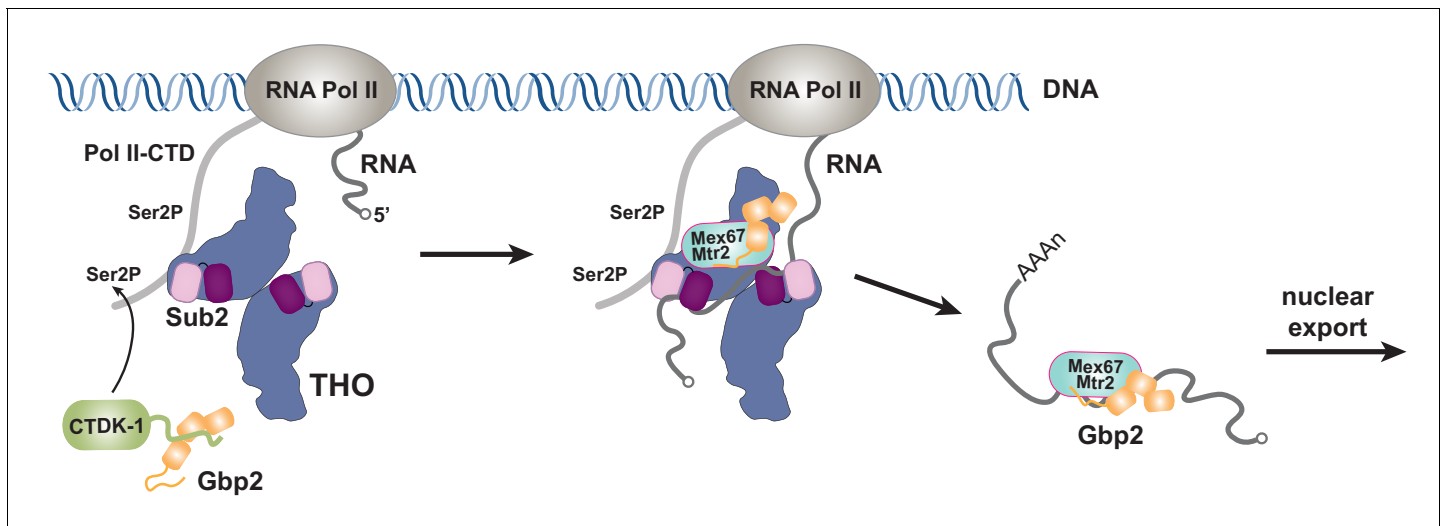

**Figure 5.** Working model of coordinated function of TREX and Gbp2. During transcription, the yeast CTDK-1 complex phosphorylates Ser2 of the RNA Pol II CTD. The N-terminal extension in CTDK-1's kinase subunit Ctk1 recognizes the RRM domains of Gbp2, connecting Gbp2 to the transcription machinery. TREX travels along with the transcription machinery and recognizes multiple domains of Gbp2, possibly facilitating its loading onto the maturing mRNP. Both TREX and Gbp2 are involved in subsequent loading of the export receptor Mex67•Mtr2 to generate export competent mRNPs.

The online version of this article includes the following figure supplement(s) for figure 5:

**Figure supplement 1.** Hypothetical model of the THO•Sub2•Yra1 complex.

speckles (*Dias et al., 2010*; *Domínguez-Sánchez et al., 2011*; *Pérez-Calero et al., 2020*; *Wang et al., 2018*). The extensive interactions between THO and Gbp2 suggest that THO could serve as a landing pad for Gbp2 loading onto mRNPs to function as a key surveillance factor during mRNP maturation.

Both Gbp2 and the TREX complex component Yra1 serve as adaptors for the export receptor Mex67•Mtr2 to facilitate its loading onto mRNPs (*Hackmann et al., 2014*; *Stutz et al., 2000*). In humans, evidence suggests that the human THO complex also makes direct contact with the export receptor (*Viphakone et al., 2012*). We previously determined the structure of Sub2 in association with RNA and the C-terminal region of Yra1 (*Ren et al., 2017*). Interestingly, Yra1 contains a second Sub2 binding site at its N-terminus (*Strässer and Hurt, 2001*; *Zenklusen et al., 2001*). It is plausible to speculate that one Yra1 can engage with two Sub2 ATPase molecules in the THO•Sub2 complex (*Figure 5—figure supplement 1*). In humans, SR proteins and Alyref, the ortholog of Yra1, bind to overlapping sites on the export receptor (*Huang et al., 2003*), suggesting that Gbp2 and Yra1 may not act together to load Mex67•Mtr2. Intriguingly, the multimeric THO complex in principle can provide a platform to engage with both Gbp2 and Yra1, with each of them recruiting one Mex67•Mtr2 molecule. Alternatively, THO may engage with Gbp2 and Yra1 during different stages of mRNP maturation. Overall, our work supports a view that THO functions as a general platform to recruit multiple export factors, but further investigations are needed to pinpoint how THO spatially and temporally coordinates the actions of different export factors to prepare mRNPs for export.

# Materials and methods

## Key resources table

| Reagent type (species) or resource | Designation | Source or reference | Identifiers | Additional information |
|---|---|---|---|---|
| Cell line (*S. frugiperda*) | Sf9 | Thermo Fisher Scientific | | |
| Cell line (*T. ni*) | High Five | Thermo Fisher Scientific | | |
| Strain (*E. coli*) | DH10Bac | Thermo Fisher Scientific | | |
| Strain (*E. coli*) | Rosetta | Novagen | | |
| Gene (*S. cerevisiae*) | Hpr1 | Uniprot | P17629 | |
| Gene (*S. cerevisiae*) | Tho2 | Uniprot | P53552 | |
| Gene (*S. cerevisiae*) | Mft1 | Uniprot | P33441 | |
| Gene (*S. cerevisiae*) | Thp2 | Uniprot | O13539 | |
| Gene (*S. cerevisiae*) | Tex1 | Uniprot | P53851 | |
| Gene (*S. bayanas*) | Tex1 | GenBank | AACA01000273.1 | Bases 2762–4030 |
| Gene (*S. cerevisiae*) | Sub2 | Uniprot | Q07478 | |
| Gene (*S. cerevisiae*) | Gbp2 | Uniprot | P25555 | |
| Software, algorithm | COOT | https://www2.mrc-lmb.cam.ac.uk/personal/pemsley/coot/ | COOT 0.8.8 | |
| Software, algorithm | Chimera | https://www.cgl.ucsf.edu/chimera/ | | |
| Software, algorithm | PyMOL | Molecular Graphics System, Schrodinger, LLC | PyMOL 2.4 | |
| Software, algorithm | Phenix | https://www.phenix-online.org | Phenix 1.11 | |

*Continued on next page*

*Continued*

| Reagent type (species) or resource | Designation | Source or reference | Identifiers | Additional information |
| --- | --- | --- | --- | --- |
| Software, algorithm | Relion | https://www3.mrc-lmb.cam.ac.uk/relion/ | Relion 3.1 | |
| Software, algorithm | pLink2 | http://pfind.ict.ac.cn/software/pLink/ | | |
| Software, algorithm | CX-Circos | http://cx-circos.net | | |
| Software, algorithm | Xlink Analyzer | https://www.embl-hamburg.de/XlinkAnalyzer/XlinkAnalyzer.html | | |

## Plasmids and proteins

Both THO–FL and the THO* core complex were expressed in High-Five insect cells by coinfection of recombinant baculoviruses. THO–FL contains full length *S. cerevisiae* Tho2 (residues 1–1597), Hpr1 (residues 1–752), Tex1 (residues 1–422), Mft1 (residues 1–392), and Thp2 (residues 1–261) subunits and the former four subunits each contains a TEV cleavable N-terminal His tag. The THO* complex contains *S. cerevisiae* Tho2 (residues 1–1257), Hpr1 (residues 1–603), Mft1 (residues 1–256), full length Thp2, and *S. bayanas* Tex1 (residues 1–380) with Tho2 and Hpr1 each containing a TEV cleavable N-terminal His tag. High-Five cells were harvested 48 hr after infection. The cells were sonicated in a lysis buffer containing 50 mM Tris pH 8.0, 300 mM NaCl, 10 mM imidazole, 1 mM PMSF, 5 mg/L aprotinin, 1 mg/L pepstatin, 1 mg/L leupeptin, and 0.5 mM TCEP. THO complexes were purified by Ni affinity chromatography, followed by TEV digestion to remove His tags. The proteins were then purified on a mono Q column (GE Healthcare) and subjected to further size exclusion purification with a Superose 6 column (GE Healthcare) in 10 mM Tris pH 8.0, 150 mM NaCl, and 0.5 mM TCEP.

GST tagged Gbp2 (residues 1–427) and Gbp2ΔRRM3 (residues 1–316) were expressed in High-Five cells. Cells were lysed in the same condition as the THO complexes. The GST tagged Gbp2 proteins were purified using glutathione sepharose 4B resin (GE Healthcare) followed by size exclusion chromatography using a Superdex 200 column (GE Healthcare) in 10 mM Tris pH 8.0, 300 mM NaCl, and 0.5 mM TCEP.

Sub2 and Gbp2ΔRS (residues 107–427) were expressed in Rosetta *E. coli* cells (Stratagene) with an N-terminal TEV cleavable GST tag. Protein expression was induced at an $OD_{600}$ of 1.0 with 0.5 mM IPTG at 20°C for 16 hr. Cells were lysed in the same lysis buffer as mentioned above. Proteins were first purified using glutathione sepharose 4B resin. For Sub2, the GST tag was removed by TEV, and the protein was purified on a mono Q column. Untagged Sub2 and GST–Gbp2ΔN were further purified on a Superdex 200 column in 10 mM Tris pH 8.0, 150 mM NaCl, and 0.5 mM TCEP.

All purified proteins were concentrated, flash frozen in liquid nitrogen, and stored at −80°C.

## Cryo-EM sample preparation and data collection

Purified THO* and Sub2 were first buffer exchanged to 10 mM HEPES pH 7.0, 100 mM potassium acetate, and 0.5 mM TCEP. THO* was incubated with threefold molar excess of Sub2 in the presence of 0.05% glutaraldehyde for 30 min at room temperature. Cross-linking was quenched with 0.1 M Tris pH 8.0 and the sample was concentrated to 0.5 mg/mL. 1.5 μL of THO*•Sub2 was applied to a glow-discharged UltrAuFoil R 1.2/1.3 grids (Quantifoil). Grids were blotted for 3 s with a blotting force of 3 and 100% humidity at 22°C and plunged into liquid ethane using an FEI Vitrobot Mark IV (Thermo Fisher).

Electron micrographs were acquired with a Titan Krios electron microscope (Thermo Fisher) equipped with a Falcon 3EC detector (Thermo Fisher). Movies were collected with EPU with a calibrated pixel size of 0.681 Å/pixel. A total of 4907 movies were collected with a defocus range from 0.8 μm to 2.0 μm. Description of the cryo-EM data collection parameters can be found in *Supplementary file 1*.

## Cryo-EM data processing

Motion correction was performed using MotionCor2 (*Zheng et al., 2017*). The parameters of the contrast transfer function (CTF) were estimated using Gctf (*Zhang, 2016*). We initially selected 396 K

particles from 4907 micrographs with automatic particle picking in RELION-3 (*Zivanov et al., 2018*). The picked particles were binned by two and subjected to reference-free 2D classification. 205 K particles were selected for 3D classification with C2 symmetry using an initial model generated by EMAN2 (*Tang et al., 2007*). Each particle contains four copies of the THO•Sub2 complex with two copies significantly more flexible than the others. 15 K particles were selected for 3D refinement using a mask covering the two ordered THO•Sub2 molecules with C2 symmetry. The particles were then re-extracted at the original pixel size of 0.681 Å/pixel and subjected to Bayesian polishing, CTF refinement, and 3D refinement. Refinement of the entire four copies of THO•Sub2 molecules generated a map at 4.80 Å resolution. We extracted 30 K THO•Sub2 protomers from the ordered two copies and refinement using a mask covering one THO•Sub2 molecule yielded a map of THO•Sub2 at 3.70 Å resolution with a sharpening B factor of 86 $Å^2$ as assessed by an FSC threshold of 0.143.

## Model building

The 3.70 Å THO•Sub2 map was used for model building in COOT (*Emsley et al., 2010*). The five subunit THO complex was built de novo. Individual RecA domains of Sub2 were placed using our previously determined atomic resolution structure (PDB ID 5SUP). The THO•Sub2 model was subjected to real-space refinement in Phenix (*Adams et al., 2010*). The final THO•Sub2 model contains Tho2 (residues 37–913, followed by 10 poly-Ala helices at the C-terminus), Hpr1 (residues 4–535), Tex1 (residues 68–366), Mft1 (residues 5–227), Thp2 (residues 8–227), and Sub2. Figures were prepared using Chimera (*Pettersen et al., 2004*) or PyMOL (Molecular Graphics System, Schrodinger, LLC).

## GST pull-down assays

1 µM of GST or GST-tagged Gbp2 variants was incubated with 1 µM of THO variants or with 1 µM of THO and Sub2 (2 µM for *Figures 4D*, 8 µM for *Figure 4—figure supplement 3*) as indicated in the binding buffer (20 mM HEPES pH 7.0, 80 mM NaCl, and 0.5 mM TCEP) at room temperature for 10 min. The reaction mixtures were then added to ~15 µL glutathione resin in an Eppendorf tube and binding was allowed to proceed at room temperature for 30 min with gentle tapping to mix every 3–5 min. Beads were washed twice with 500 µL washing buffer containing 20 mM HEPES pH 7.0, 80 mM NaCl (for *Figure 1B*) or 50 mM NaCl (for *Figure 1C*, *Figure 4D*, and *Figure 4—figure supplement 3*), and 0.5 mM TCEP. Bound proteins were eluted with washing buffer supplemented with 20 mM glutathione and analyzed using Coomassie-stained SDS-PAGE gels. The experiments were repeated three times independently.

## Cross-linking mass spectrometry analysis

For EDC cross-linking, 1 µM of THO–FL and 1 µM of GST–Gbp2 were incubated in 10 mM HEPES pH 7.0, 105 mM NaCl, 0.5 mM TCEP in the presence of 20 mM EDC and 0.5 mM sulfo-DHS at room temperature for 40 min. The reaction was quenched at room temperature for 20 min by adding Tris pH 8.0 and β-mercaptoethanol to a final concentration of 50 mM and 20 mM, respectively. DSS cross-linking was performed in the same conditions except that 0.5 mM DSS was used and only Tris pH 8.0 was used for quenching the reaction.

The DSS and EDC cross-linked samples were directly processed for in-solution Trypsin and Lys-C digestion. The samples were reduced with 5 mM DTT and 5 mM TCEP in 8 M urea buffer (50 mM ammonium bicarbonate) and were then incubated with 30 mM iodoacetamide at room temperature for 30 min in the dark. 30–45 µg of the purified complex was digested with Trypsin and Lys-C using a 1:100 ratio for each protease upon diluting the sample to 1 M urea. The proteolysis reaction occurred overnight (12–16 hr) at 37˚C. After overnight digestion with trypsin, the complex was digested with an additional 1:100 ratio of trypsin at 37˚C for 2 hr. The resulting mixture was acidified and desalted by using a C18 cartridge (Sep-Pak, Waters).

1–2 µg of the trypsin digested cross-linked complex was analyzed with a nano-LC 1200 that is coupled online with a Q Exactive HF-X Hybrid Quadrupole Orbitrap mass spectrometer (Thermo Fisher) (*Xiang et al., 2020a*; *Xiang et al., 2020b*). Briefly, desalted peptides were loaded onto a Picochip column (C18, 1.9 µm particle size, 200 Å pore size, 50 µm × 25 cm; New Objective) and eluted using a 60 min liquid chromatography gradient (5% B–8% B, 0–2 min; 8% B–40% B, 2–50 min; 40%B–100% B, 50–60 min; mobile phase A consisted of 0.1% formic acid (FA), and mobile phase B

consisted of 0.1% FA in 80% acetonitrile). The flow rate was ~350 nL/min. The QE HF-X instrument was operated in the data-dependent mode, where the top six most abundant ions (mass range 350–2000, charge state 4–8) were fragmented by high-energy collisional dissociation (HCD). The target resolution was 120,000 for MS and 15,000 for tandem MS (MS/MS) analyses. The quadrupole isolation window was 1.6 Th, and the maximum injection time for MS/MS was set at 300 ms.

After the MS analysis, the data was searched by pLink2 for the identification of cross-linked peptides (*Chen et al., 2019*). The mass accuracy was specified as 10 and 20 p.p.m. for MS and MS/MS, respectively. Other search parameters included cysteine carbamidomethylation as a fixed modification and methionine oxidation as a variable modification. A maximum of three trypsin missed-cleavage sites were allowed. The cross-link spectra were then manually checked to remove potential false-positive identifications as previously described (*Xiang et al., 2020b*). The cross-linking data was analyzed by CX-Circos (http://cx-circos.net). The distance distribution of the cross-links onto the THO structure was performed with Xlink Analyzer (*Kosinski et al., 2015*).

## Acknowledgements

We thank Scott Collier and Melissa Chambers at the Center for Structural Biology Cryo-EM Facility at Vanderbilt University for assistance in Cryo-EM data collection. We thank members of the Wente laboratory for discussions. This work was supported by NIGMS grants R35 GM133743 to YR and GM137905 to YS, and funds from Vanderbilt University School of Medicine to YR. BPC was supported by NIH/NCI training grant T32CA119925.

## Additional information

### Funding

| Funder | Grant reference number | Author |
| --- | --- | --- |
| National Institute of General Medical Sciences | GM133743 | Yihu Xie<br>Bradley P Clarke<br>Austin L Ivey<br>Pate S Hill<br>Yi Ren |
| National Institute of General Medical Sciences | GM137905 | Yong Joon Kim<br>Yi Shi |
| National Cancer Institute | T32CA119925 | Bradley P Clarke |
| Vanderbilt University School of Medicine | | Yi Ren |

The funders had no role in study design, data collection and interpretation, or the decision to submit the work for publication.

### Author contributions

Yihu Xie, Conceptualization, Data curation, Formal analysis, Writing - original draft, Writing - review and editing; Bradley P Clarke, Formal analysis, Writing - original draft, Writing - review and editing; Yong Joon Kim, Data curation, Formal analysis, Writing - original draft, Writing - review and editing; Austin L Ivey, Pate S Hill, Data curation; Yi Shi, Formal analysis, Supervision, Funding acquisition, Writing - original draft, Writing - review and editing; Yi Ren, Conceptualization, Data curation, Formal analysis, Supervision, Funding acquisition, Writing - original draft, Writing - review and editing

### Author ORCIDs

Bradley P Clarke  http://orcid.org/0000-0002-9413-9905
Pate S Hill  http://orcid.org/0000-0001-9550-2713
Yi Shi  http://orcid.org/0000-0002-2761-8324
Yi Ren  https://orcid.org/0000-0003-4531-0910

**Decision letter and Author response**
Decision letter https://doi.org/10.7554/eLife.65699.sa1
Author response https://doi.org/10.7554/eLife.65699.sa2

## Additional files

### Supplementary files
• Supplementary file 1. Cryo-EM data collection, refinement, and validation statistics.

• Supplementary file 2. Unique EDC and DSS cross-linked peptides identified from the THO–Gbp2 complex.

• Transparent reporting form

### Data availability

The cryo-EM density maps have been deposited in the Electron Microscopy Data Bank under the accession number EMD-23527. The coordinates of the THO•Sub2 complex has be deposited in the Protein Data Bank under the accession number 7LUV.

The following datasets were generated:

| Author(s) | Year | Dataset title | Dataset URL | Database and Identifier |
|---|---|---|---|---|
| Xie Y, Ren Y | 2021 | Cryo-EM structure of the yeast THO-Sub2 complex | https://www.ebi.ac.uk/pdbe/entry/emdb/EMD-23527 | Electron Microscopy Data Bank, EMD-23527 |
| Xie Y, Ren Y | 2021 | Cryo-EM structure of the yeast THO-Sub2 complex | https://www.rcsb.org/structure/7LUV | RCSB Protein Data Bank, 7LUV |

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
