## [Decision Letter]

**Acceptance summary:**

This manuscript describes the structure of the yeast THO:Sub2 complex and how it interacts with the SR like protein Gbp2. The paper extends two recently published Tho:Sub2 complex structures by the Conti and Plaschka groups in two ways. Firstly, it shows how Gbp2 interacts with the THO complex. Secondly, it reveals a substantially different orientation between THO:Sub2 protomers compared with the earlier structure, so provides more information on the flexibility and range of movements that the two protomers might make with respect to each other.

**Decision letter after peer review:**

Thank you for submitting your article "Cryo-EM structure of the yeast TREX complex and coordination with the SR-like protein Gbp2" for consideration by *eLife*. Your article has been reviewed by 2 peer reviewers, one of whom is a member of our Board of Reviewing Editors, and the evaluation has been overseen by Cynthia Wolberger as the Senior Editor. The reviewers have opted to remain anonymous.

Essential Revisions:

The two reviewers make several constructive suggestions to improve the presentation of the manuscript. The requested edits are clearly explained in the reviews that are attached below and it should be straightforward to implement all those suggestions.

Reviewer #1 (Public Review):

1. I found the initial description of the overall structure confusing. At first the authors say the complex is a tetramer, which is not what was seen by the Conti lab and then follow that with a confusing discussion leading to the conclusion that the dimer with a rigid subunit and a flexible one is the functional unit. It rather feels like they arrive at this conclusion because that's what Conti's lab saw, rather than any other reason. Since the human complex is a tetramer, perhaps the tetrameric complex observed here is one possible form and that possibility should be considered more carefully. Please state whether there is any similarity in the arrangements between the human tetramer and the tetramer observed here. I found the figure 2 supp 1C was not easy to follow. Coloring each of the four protomers differently would make things clearer.

2. The authors previously determined the structure of yra1C domain bound to sub2 and several labs have shown this interaction activates Sub2 atpase activity. Are those interaction observed previously between Yra1 and Sub2 compatible with this new structure? If so, perhaps the authors could provide a model showing how Yra1 fits into this larger complex. Also, could Yra1 C domain and Gbp2 bind simultaneously to a single THO-Sub2 protomer or would one protomer bind Yra1 and perhaps another bind Gbp2? This is worth considering because this would strengthen the concept that TREX acts as a general platform engaging with multiple export factors to drive recruitment of multiple Mex67 molecules and eventual export of the Mex67:mRNP complex. In the human system, the SR proteins and Alyref have an overlapping binding site on Nxf1, suggesting they may not act together to recruit a single Nxf1, but rather they recruit different Nxf1 molecules perhaps to the same mRNP via a single multimeric THO platform.

Reviewer #2 (Public Review):

Overall this is a solid and technically sound manuscript, and I have only two relatively minor suggestions for improvement.

1. Tetramer versus dimer

The particles that were analyzed by cryoEM were composed of four THO-Sub2 protomers, yet the authors argue that a dimer is the functional unit. Why? The tetramer versus dimer organization needs to be better discussed, also in light of the observation that the human complex can also form a tetramer.

2. Sub2 activation mechanism

The authors should more carefully discuss how THO 'activates' Sub2 (and how the 'semi-open state' leads to activation) and indicate the RNA binding surface of Sub2 in their model.

---

## [Author Response]

Reviewer #1 (Public Review):1. I found the initial description of the overall structure confusing. At first the authors say the complex is a tetramer, which is not what was seen by the Conti lab and then follow that with a confusing discussion leading to the conclusion that the dimer with a rigid subunit and a flexible one is the functional unit. It rather feels like they arrive at this conclusion because that's what Conti's lab saw, rather than any other reason. Since the human complex is a tetramer, perhaps the tetrameric complex observed here is one possible form and that possibility should be considered more carefully. Please state whether there is any similarity in the arrangements between the human tetramer and the tetramer observed here. I found the figure 2 supp 1C was not easy to follow. Coloring each of the four protomers differently would make things clearer.

We thank the reviewer for this comment. We rewrite the description of the THO•Sub2 tetramer and include a new figure (Figure 2-figure supplement 2) in this revision. We think the THO•Sub2 dimer is a relevant assembly based on the following reasons: (1) Full length THO isolated from yeast cells exhibits a dimer (Pena et al, EMBO J 2012). (2) Our XL-MS analysis of full length THO (in the presence of Gbp2) provides experimental evidence that supports the dimer, whereas no crosslinks were found to support the tetramer configuration. (3) The recently reported structure of full length THO bound to Sub2 by the Conti lab exhibits a dimer (Schuller et al, eLife 2020). (4) Tetramerization of human THOUAP56 (Puhringer et al, eLife 2020) is mediated by THO components (Thoc6 subunit and Thoc5 residues 250-683) that do not exist in yeast THO.

As the THO* core complex used in our cryo-EM work contains ~600 residues less than the full length THO, it is possible that the tetramer was formed due to the truncations in THO*. No other evidence thus far supports the physiological relevance of the tetramer. Given the multiple independent lines of evidence supporting the dimer configuration, we chose to focus on the dimer in this study.

2. The authors previously determined the structure of yra1C domain bound to sub2 and several labs have shown this interaction activates Sub2 atpase activity. Are those interaction observed previously between Yra1 and Sub2 compatible with this new structure? If so, perhaps the authors could provide a model showing how Yra1 fits into this larger complex. Also, could Yra1 C domain and Gbp2 bind simultaneously to a single THO-Sub2 protomer or would one protomer bind Yra1 and perhaps another bind Gbp2? This is worth considering because this would strengthen the concept that TREX acts as a general platform engaging with multiple export factors to drive recruitment of multiple Mex67 molecules and eventual export of the Mex67:mRNP complex. In the human system, the SR proteins and Alyref have an overlapping binding site on Nxf1, suggesting they may not act together to recruit a single Nxf1, but rather they recruit different Nxf1 molecules perhaps to the same mRNP via a single multimeric THO platform.

We have included a new figure (Figure 5-figure supplement 1) to illustrate a hypothetical model of the THO•Sub2•Yra1 complex with Sub2 in an RNA-bound closed conformation. The Sub2-Yra1 interaction observed in our previous crystallization studies (Ren et al, eLife 2017) is compatible with the THO•Sub2 cryo-EM structure. A potential assembly containing one copy of Yra1 and two copies of THO•Sub2 is depicted, which is also suggested by the recent work on yeast THO•Sub2 and human THO•UAP56 (Schuller et al, eLife 2020; Puhringer et al, eLife 2020). Our model also suggests that the RNA binding regions of Yra1 (RRM domain and the regions flanking RRM) are placed in close proximity to the RNA substrate.

As the reviewer points out, how THO coordinates the actions of different export factors including Yra1 and Gbp2 to load Mex67•Mtr2 is an intriguing question. We add discussion on this point in the revised manuscript. We agree with the reviewer that the multimeric nature of the THO complex provides a possibility that THO can coordinate the actions of Yra1 and Gbp2 simultaneously in one complex, thereby recruiting multiple export receptor molecules. On the other hand, Yra1 and Gbp2 binding to THO may occur in different settings during mRNP maturation. Further investigations are needed to differentiate these two possibilities.

Reviewer #2 (Public Review):Overall this is a solid and technically sound manuscript, and I have only two relatively minor suggestions for improvement.1. Tetramer versus dimerThe particles that were analyzed by cryoEM were composed of four THO-Sub2 protomers, yet the authors argue that a dimer is the functional unit. Why? The tetramer versus dimer organization needs to be better discussed, also in light of the observation that the human complex can also form a tetramer.

We thank the reviewer for the comment. As mentioned above in response to Reviewer 1’s comment, we rewrite the description of the THO•Sub2 tetramer and include a new figure (Figure 2—figure supplement 2) to address this point. Briefly, our work (by XL-MS) and studies by others (Pena et al., EMBO J 2012; Schuller et al., *eLife* 2020) provide strong experimental evidence to support the relevance of the dimer, whereas no evidence thus far supports the tetramer configuration. Therefore, we focus on the dimer in this study.

2. Sub2 activation mechanismThe authors should more carefully discuss how THO 'activates' Sub2 (and how the 'semi-open state' leads to activation) and indicate the RNA binding surface of Sub2 in their model.

We now add discussion about the Sub2 regulation mechanism. Sub2 undergoes different conformational states during the enzymatic cycle. We think the semi-open Sub2 stabilized by THO would change the conformational dynamics of Sub2 in a way that enables efficient transition between different states. in vivo, the role of this regulation mechanism is two-fold. First, THO would recruit Sub2 to the transcription machinery so that Sub2 only acts on its physiological RNA substrate. Second, THO stabilizes Sub2 in a primed state to allow efficient engagement with the substrate.

The RNA binding surface of Sub2 is on the opposite side to the THO binding site. In the newly added hypothetical model of THO•Sub2•Yra1 (Figure 5—figure supplement 1), we illustrate the potential path of an RNA substrate. The configuration of the two Sub2 molecules would allow THO•Sub2 to engage with a longer stretch of RNA. We have also edited Figure 5 to include Sub2 and RNA.